# Enhancing the Understanding of the EU Gender Equality Index through Spatiotemporal Visualizations

**Laya Targa** [1],*, **Silvia Rueda** [2], **Jose Vicente Riera** [1], **Sergio Casas** [1] **and Cristina Portalés** [1]

1 Institute of Robotics and Information and Communication Technologies, Universitat de València, 46980 Paterna, Spain
2 Computer Science Department, Universitat de València, 46100 Burjassot, Spain
* Correspondence: laya.targa@uv.es

**Abstract:** The Gender Equality Index allows analyzing and measuring the progress of gender equality in the EU and, therefore, the relation between men and women in different domains, such as Health, Work or Money. Even though the European Institute for Gender Equality has created some visualizations that are useful to look at the data, this website does not manage to make graphs that allow for observing the spatiotemporal variable. This article enhances the understanding of the index with spatiotemporal visualizations, such as cartograms, heatmaps and choropleth maps, and some strategies focusing on analyzing the evolution of the countries over the years in an open-access environment. The results show how the application created may be used as an addition to the EIGE website.

**Keywords:** data visualization; cartogram; choropleth map; heatmap; interaction; gender equality

## 1. Introduction

Gender Equality (GE) has been one of the main concerns of the European Union (EU) since its inception and has been addressed from different perspectives that have marked the evolution of equality as a superior community value.

The EU has a long history of developing public policies on GE, whose origin lies in the 1957 Treaty of Rome (which enshrined equal pay). Since then, the EU has adopted numerous directives in the field of gender equality, for example, those on equal pay and social security, the protection of pregnant women and people on parental leave, as well as access to goods and services.

One of the latest initiatives inside the European Council is the European Research Area (ERA) Policy Agenda [1]. The ERA Policy Agenda sets out voluntary ERA actions for the period 2022–2024 to contribute to the priority areas defined in the Council Recommendation on a Pact for Research and Innovation in Europe (Pact for R&I). The list of actions draws mainly on the Commission's Communication "A new ERA for Research and Innovation" of September 2020 and on the Council conclusions of December 2020 on the "New European Research Area". It also takes into account the Council conclusions on "Deepening the ERA: Providing researchers with attractive and sustainable careers and working conditions and making brain circulation a reality" of May 2021 and on the "Global approach to Research and Innovation" of September 2021. In particular, as Gender Equality is a core EU value and gender mainstreaming a core EU strategy, the ERA Action 5, "Promote gender equality and foster inclusiveness, taking note of the Ljubljana declaration", is defined to promote an institutional change in R&D organizations that allows achieving sustainable and long-term progress towards gender equality in R&I.

Among the initiatives undertaken by the European Union, with the aim of promoting and reinforcing equality between men and women, it is worth mentioning the creation, in 2006, of the European Institute for Gender Equality (EIGE) [2], which carries out fundamental research work on Gender Equality and the Gender Gap.

The Gender Gap is an analytical and empirical construction that arises from the difference between the categories of a variable in relation to male and female rates [3]. It highlights the existing inequalities between men and women in any field, in relation to the level of participation; access to opportunities, rights, power and influence; remuneration and benefits; control; and use of resources, which allow them to guarantee their well-being and human development. Gender gaps are expressed in all areas of performance, such as economic, social, labor, cultural, health, etc. They are based on the hierarchy of the differences between men and women and are expressed in different ways, depending on the area in question.

One of the most important areas of study of the EIGE is, precisely, the development of the Gender Equality Index [4], a powerful tool to measure the progress of gender equality in the European Union (EU). This index, elaborated since 2013, includes data from all EU countries about 31 different indicators in six different domains (work, money, knowledge, time, power and health) measuring gender equality to provide effective information for policymakers to design more effective actions focused on increasing gender equality. This index has been reported to be an integral part of effective policymaking in the EU [5], and even acknowledged as the most complete and detailed gender equality indicator [6], also proposing similar indicators at a regional level in Italy [6] or Great Britain [7]. This index is used as a basis for other studies, for instance, to show that promoting gender equality contributes towards narrowing the magnitude of the differences in political interest between men and women [8]. On the other hand, others point out some limitations of the index, such as the lack of transparency around the methodological decisions or that it predominantly captures achievement levels rather than gender gaps [9,10]. Also, in [11], it is reported the fact that conscription (compulsory military service) is not considered in the index.

To really have a real impact on reducing inequalities, it is essential that collected data be correct and adjusted, but also that it can be easily interpreted; otherwise, it would be very difficult to define the most appropriate policies. For this purpose, it is mandatory to provide an adequate and simple way to visualize data, especially when we are considering huge amounts of data, as in this case.

But, in the midst of a global crisis of conflict within the EU's own borders, in the aftermath of a global pandemic, with an inescapable climate emergency, serious problems of large-scale conflict and displacement, and huge political crises, progress towards gender parity continues to fall short of goals. As leaders grapple with an increasing series of economic and political shocks, the risk of a reversal intensifies. Not only are millions of women and girls losing access and opportunities today, but this disruption of progress towards parity is a catastrophe for the future of our economies, societies and communities. Accelerating parity is, therefore, a central part of the European agenda. Thus, it is urgent to define truly effective action policies.

If we really want our politicians to be able to take measures based on evidence, it is necessary that the data be presented in a very clear way, providing the essential temporal vision that allows us to understand if the measures that have been applied in the different countries of the EU have had an effect; if a certain line of action must be continued; or if, on the contrary, it is necessary to leave behind old ideas and face the situation with new measures.

Going back to the case of the data provided by the EIGE to compute the Gender Equality Index, the visualizations used to measure the complex concept of gender equality and to monitor the progress of gender equality across the EU over time are mainly composed of an interactive radar chart that summarizes the GEI indicator for each country and for the EU, as shown in Figure 1 (from [12]). When clicking in one of the countries, pie charts for different domains (Work, Money, Health, etc.) are depicted, and when clicking on one of the pie charts, detailed information on the related domain is given and complemented with bar charts and tables.

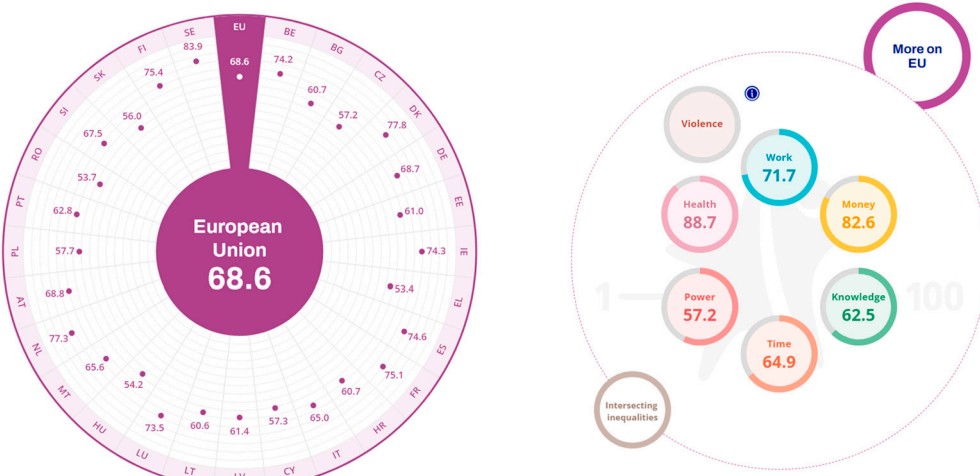

**Figure 1.** A snapshot taken from the EIGE's webpage, where the GEI for the year 2022 is shown, with the focus on the values for the European Union.

Even though these graphs are useful to provide quick and synthesized access to the values of the GEI indicators and domains for each year and country, they present some problems when trying to identify trends and topological relationships, because the spatial and temporal dimensions are not represented simultaneously, as stated in [13].

In order to allow our politicians to be truly capable of taking evidence-based measures, we propose to improve the understanding of the GEI by expanding the visualizations provided in [12], presenting them much more clearly and providing the essential temporal vision that allows for understanding if the measures that have been applied in the different EU countries have had an effect; if a certain line of action should be continued; or, on the contrary, it is necessary to leave behind old ideas and face the situation with new measures.

Moreover, the amount of data currently generated or used in research carried out with funds, mostly public, thanks to European funds, is growing exponentially. Unfortunately, a significant part of these data never becomes usable by other researchers. Hence, there is a growing concern in the EU about the need for open and reusable scientific data and other digital research results. This justifies the implementation of the European Open Science Cloud (EOSC) [14] and the inclusion of the Action 1, "Enable the open sharing of knowledge and the re-use of research outputs, including through the development of the European Open Science Cloud (EOSC)", in the ERA Policy Agenda [1].

Considering this as a mandatory recommendation, we state that the proposed results should be open and accessible not only to the governments of the different countries, but also to all the research teams that work internationally to help close the gender gap. For that reason, we have developed an interactive webpage where these data can be viewed openly.

The rest of this paper is organized as follows. In Section 2, some related work is commented on. In Section 3, details on data processing are given, first explaining the GEI; then the available visualizations; and, finally, the new strategies developed in the current work. In Section 4, we provide details on the design and implementation. Section 5 shows some results, based on two case studies where the new graphs provided in the current work are compared to the ones provided by EIGE. Finally, a discussion and conclusion section is provided.

## 2. Related Work

There is a big difference between having data and having useful information. Beyond the need to have adequate and unbiased data that represent an entire population, which, in fact, has been really difficult over the years when talking about gender bias [15], many different factors intervene for these data to become useful information.

Working with large volumes of data can be a challenge for some people. That is why data visualization is an effective way to understand complex data, which happens to be super essential for many fields. But, not all ways of visualizing data are equally effective.

Visual stimuli have a profound impact on our cognitive processes and understanding of information. Our brain is very aware of visual cues and patterns. As a result, our visual perception is one of our fundamental abilities to interpret and make sense of the world around us [16]. The human brain is designed to process visual stimuli more effectively than numerical data. Presenting data visually means taking advantage of this innate human ability to recognize patterns, identify relationships and extract information [17]. Understanding how visual perception works and what impact it has allows us to create visualizations that are easier to remember and thus facilitate decision making [18]. In that sense, it can be said that Data Visualization lets viewers see beyond summary statistics.

Data visualization provides intuitive ways to interactively explore and analyze massive datasets, as a means to transform huge quantities of raw data into graphical representations and images that can be quickly decodified by our brains. And it is particularly useful when these data are dynamic, noisy and/or heterogeneous, enabling us to effectively identify interesting patterns, infer correlations and causalities, and support sense-making activities [19], making it possible to amplify human cognition [20,21]. Moreover, in addition to visualizing and analyzing the message contained in the data, graphical visualization of data also makes possible a longer memory persistence, because it has been reported that most people have a more persistent visual memory than verbal or auditory [22]. Data visualization has thus become one of the cornerstones of data science, turning the abundance of Big Data produced through modern systems into actionable knowledge [23], making it possible to synthesize tons of data into visual forms that humans can understand.

The way in which we process the information that our brain receives and how visual information is presented affects the understanding of the information, and the decision-making process has been deeply studied from Human Psychology [17], proving that there are many different factors that need to be addressed in order to develop effective and impactful visualizations for all individuals. This does not include only aesthetics, like considering the graph colors to make them color-blind friendly, or the need of interactivity to provide a spatiotemporal context [24], but also considers equity and diversity from an intersectional point of view to guarantee not only that the information is accessible by everyone, but also that it takes into account the objectives and problems of all societies [25].

It is worth highlighting the relevance of data visualization for decision making, a practice quite old, as decisions have been traditionally supported by maps and other charts [26–28]. Moreover, Data Visualization has been applied to fields as diverse as Marketing [29], Health and Social Care [30] or Politics [31,32].

In fact, one of the most impactful examples of data visualization was during the 2020 pandemic. It was key in the efforts to communicate the science around COVID-19 to the wide audience of policymakers, scientists, professionals in health and the general public, helping to understand the different aspects of the pandemic [33].

Data Visualization has traditionally been used in gender studies to highlight gaps and reveal differences in the situations experienced by women and men around the world in an infinite variety of aspects. Over the last few years, a huge variety of studies and applications have emerged to help the general or specific public to understand big data information. For instance, in [34], data visualization was used to explain to students the gender pay gap. In [35], the authors provide a discussion on gender bias in the news using their own visualization framework. In [36], the authors show how visualization can prove the relationship among gender, race, citizenship and academic performance. Or in [37], it is discussed how visualization can be used for making teachers realize racial and gender inequity patterns in student engagement in their classrooms.

In the specific case of Gender Gap studies, in recent years there have been numerous examples of interactive webpages that have applied visualization of massive data to this area. This can be seen, for instance, on the website Flowingdata [38], an independent site

where people can share their graphics, where a good part of these projects are labeled as gender related [39]. Among the different projects, we can find a study [40], that, using a line graph, shows the decrease in the presence of women in computer science over the last 30 years, contrary to what happened in other technical fields. Another example studies the gender gap in PhDs around the world [41], using a bubble plot, and shows that, in almost three-quarters of the 56 nations considered, more men than women are awarded a PhD. Finally, it is worth noting the World Bank project [42], because the relevance of data visualization to identifying the gender gap is explicitly recognized. The World Bank updated its Gender Data Portal with different visualizations in an effort to make gender inequalities more obvious, which accompanied the following comment: "The World Bank Group has redesigned its Gender Data Portal with these audiences in mind by offering over 900 gender indicators in different formats, ranging from raw data to appealing visualizations and stories. Making sex-disaggregated data easier to analyze, interpret and visualize will bring into focus gender issues that are frequently invisible, including on topics such as digital development, transport and water. It will highlight existing gender gaps as well as gaps in the availability of gender data".

As it can be seen from the previous works, the richness of data visualization and interactive graphs is used in many ways to provide added value to raw data. Either one or the other graphs are used to transmit such values, as it is well known that some graphs work better for certain purposes (e.g., barplots are useful to show rankings), although they are not free of caveats (e.g., too many line plots are difficult to understand) that have to be avoided [43,44]. Our proposed visualizations take in consideration benefits of different graphs for the purpose of enriching the EIGE's visualizations by taking in consideration the spatial and temporal dimension of data, as addressed in the following sections.

## 3. Data Processing

### 3.1. Gender Equality Index

The Gender Equality Index (GEI) uses different factors to determine the relations between men and woman in six domains (Work, Money, Knowledge, Time, Power, Health). This index is measured on a scale of 1 to 100, where a higher value represents higher equality between men and women.

The European Institute for Gender Equality used an approach to obtain the GEI based on weights, aggregation, normalization and imputation of the domains and sub-domains, which is detailed in the methodological report described in [45]. Equation (1) indicates how the GEI is calculated according to [45]; this equation shows the importance, and therefore the weight, of each domain to calculate the indicator.

$$\text{GEI} = \text{Work}^{0.19} \times \text{Money}^{0.15} \times \text{Knowledge}^{0.22} \times \text{Time}^{0.15} \times \text{Power}^{0.19} \times \text{Health}^{0.1} \quad (1)$$

The dataset with the values of each domain and the GEI is accessible at [46] for a set of years (2013, 2015, 2017, 2019, 2020, 2021 and 2022), in an Excel file. It has to be noted that, despite having data with a temporal variable (index year), the raw data are based on prior years (2010, 2012, 2015, 2017, 2018, 2019 and 2020, respectively).

The information provided in the dataset is broken down into the score for each category of each domain. There are also some indicators, such as the career prospects index or share of ministers for each gender (man and woman), and the total. For this paper, a new simplified dataset was created based on the raw data, but using only the values of the main domains and the GEI calculated for each year, which are the ones that we use for the proposed visualizations.

As previously stated, the interactive visualizations proposed by EIGE can be enhanced, and that is the main objective of this work. Our working methodology is based on (i) analyzing existing visualization solutions proposed by EIGE; (ii) proposing improvements and new visualization strategies; (iii) designing and implementing such solutions, first collecting the raw data and finally creating open solutions; and (iv) analyzing the functioning of these methods with specific cases, including singularities, such as visualizations of small

countries. Each of these phases is developed in the following Sections 3.2, 3.3, 4 and 5 of the article.

### 3.2. Interactive Visualizations Proposed by EIGE

The EIGE has proposed some interactive visualizations to help users understand the raw data that can be accessed through [12]. The most visible graph—accessible on the main webpage—is a radar chart (Figure 1), used to show the GEI index of each country and the average value of the EU. When a user clicks in one of the countries, the GEI value appears at the center of the radar chart, while the values of the related domains appear on the right side, visualized as embedded in a doughnut chart. For instance, the radar chart in Figure 1 shows in its center the GEI value for the European Union (calculated as the average of all countries) for the year 2022, which is 68.6. The individual values for the rest of the countries are also displayed in the radar chart, although not visualized in its center, but arranged in the radial axes with dots.

This type of visualization looks appealing, is easy to understand, and minimal efforts are required to interact with the graph, i.e., selecting a year on a dropdown menu and selecting a country either in a dropdown menu or directly from the chart.

Radar charts are widely used to represent scientific data, and some other examples can be found related to gender inequality, e.g., to represent gendered shared resources [47], to evidence gender disparity [48] in tourism management or to map vulnerability indices [49], to name some. However, while radar charts (also known as spider charts) can be used to display numerical information about categorical data, they are reported by some authors to bring a few disadvantages [50–52]. One of the main problems is that using a circular layout as an axis makes the values difficult to read. Additionally, when using multiple categories, depending on which order the author follows to display them on the charts, the charts can create different shapes that can be confusing. In relation to the current research, we can see that this chart lacks the geographical dimension, and it is not easy to compare the evolution of such indicators over time, because only one year is depicted at a time.

Apart from the radar chart provided at the main webpage, the EIGE provides other types of graphs to compare the values between different countries. For instance, in the "compare countries" tab, data can be displayed either in the form of a map, a radar chart or a barplot or in a tabular way. For instance, Figure 2 shows the barplot representation for the year 2022; it is sorted by the GEI value of each country, with an accent color for the mean value of the European Union (the darker bar with the label EU). This representation is very useful to see the rankings, which reveals which countries have the best (SE, NK and NL, in Figure 2) and worst (HU, RO and EL, in Figure 2) values, as well as which countries are above or below the mean. On the other hand, this graph lacks interactivity and has similar flaws as radar charts in relation to the spatial and temporal dimensions.

There is also a section that helps users analyzing the index and the indicators of a selected country, accessible through the "view countries" tab. The graphs used in this case are doughnuts, barplots and line plots (composed of scatter plots with connected dotted lines). For example, Figure 3 exhibits the information related to Spain in a line plot, using different colors to distinguish between each domain. Users can click on a given domain to activate/deactivate its visualization.

This graph allows visualizing the temporal evolution of each domain for the given country, which is of interest for the current work. However, such trends cannot be easily compared with other countries, as this would require selecting one country each time, without the possibility to compare all at the same time. It must be noted that using this type of visualization on a selected country is an excellent choice because it is clean, as only seven categories are displayed (and, thus, seven lines). However, such a graphical representation of all the countries would lead to the so called "spaghetti plot" [53], as it would have $7 \times 28$ lines, which would make the plot unreadable. An alternative solution to represent many lines while avoiding this caveat could be to include interactivity in the graph, in such a way that one or another line is highlighted according to the user

navigation on top of the graph. However, while this solution would make it easy to read a single line, the comparison with the rest could be difficult. Thus, other types of visualizations can be explored, such as the heatmaps, as proposed in this paper. With such representations, we will tackle the issue of easily comparing the temporal evolution of the GEI and the related domains for all countries, but other graphs will be explored that include the geographical dimension.

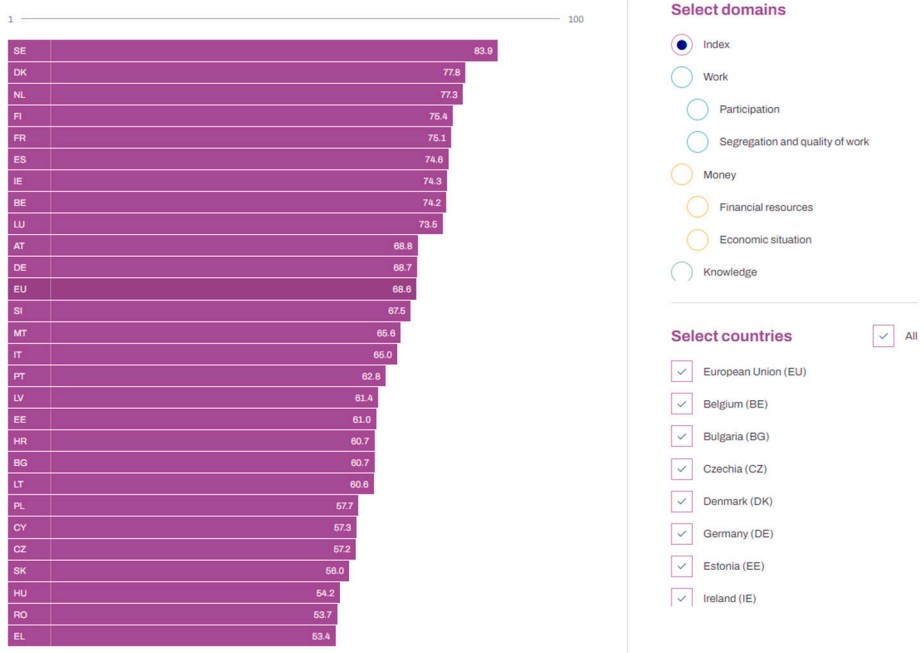

**Figure 2.** A snapshot taken from the EIGE's webpage, where the GEI for the year 2022 is shown, allowing the user to compare the values of different countries.

### Trends in Spain

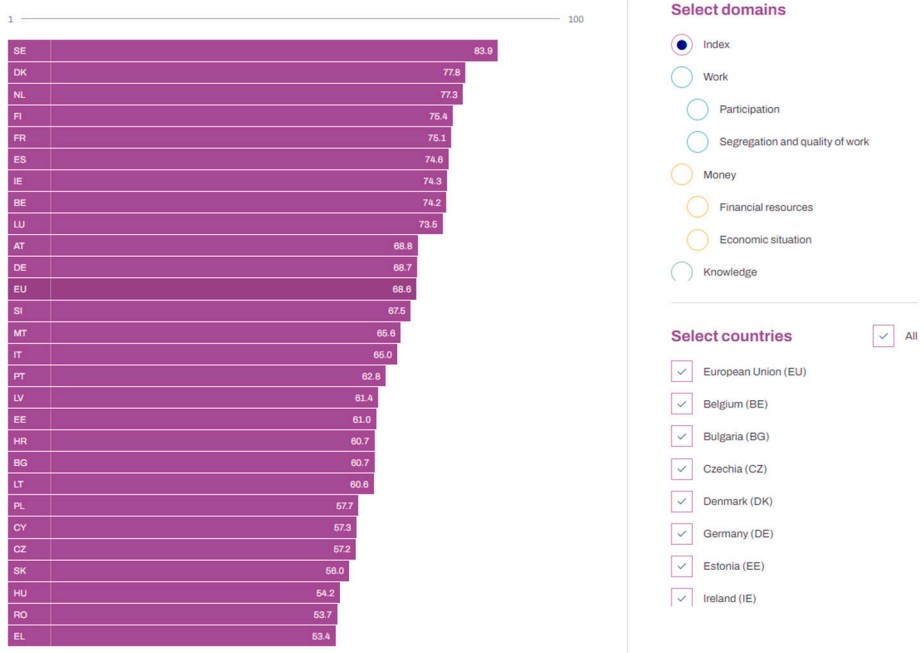

**Figure 3.** A snapshot from the EIGE's webpage, with the evolution of Spain GEI and domains, shown as a line plot.

To sum up, the overall experience when using the EIGE's website to visualize the GEI data is great, as different interactive visualizations are available, with the possibility to choose between years and different countries. However, as seen by the given examples, such visualizations lack a simple comparison when it comes to combining space and/or time. For instance, there are dropdowns that can be used to filter the information by a specific year, but there is no possible way to compare different time periods side by side or to know how much a country has evolved over the years in comparison with the rest of the countries.

### 3.3. Enhanced Proposed Visualizations

To improve the visualizations provided by EIGE, we have implemented a set of visualizations that consider the spatial and/or the temporal dimension. Additionally, we implement some strategies to make the comparison between different years, countries and indicators easy. These are reported in the following sub-sections.

### 3.3.1. Types of Graphs

Regarding the types of graphs to enhance the GEI visualization, we have used maps, cartograms and heatmaps:

- Maps: Maps are the most used visualizations when dealing with geographic data. Sometimes those maps have as background base maps from providers, such as OpenStreetMap, Carto or Esri, and other times there is only a shapefile that shows the state or country borders. Using the aesthetic color to map a certain variable on a given country leads to choropleth maps, which is the proposal of this work. Using this type of visualization ensures that the topological relations of the countries (e.g., what countries are to the west of a given one) and their sizes are maintained. It also helps in easily identifying patterns by looking at the colors that are filling the surface of each country. However, from merely looking at the color, it is not straightforward to determine the exact value. Therefore, we have added interactivity to the maps, so the numerical value is displayed when a user clicks on one of the countries. An example of this graph is shown in Figure 4a, where the label for Spain is seen.
- Cartograms: Cartograms are types of maps in which the size of the regions has been altered or modified according to the value of a variable. Since we wanted to avoid huge distortion of the regions, while compensating for the different sizes of the countries (e.g., Malta has 316 km$^2$, Spain has 506,030 km$^2$), we have used a specific type of cartogram, which is usually referred to as "cartogram heatmap" [43], "tile maps" [54] or "equal area unit map" [55]. In this type of graph, all the countries are represented by the same shape (usually rectangles, squares, triangles or hexagons), and a color ramp is used to map a certain variable to the shape. An example of this graph is shown in Figure 4b, where the countries are displayed; the label of Spain is shown, as well as the value of GEI and the corresponding year.
- Heatmap: The heatmap is a visualization that represents the values in a matrix, typically on a square matrix, but not necessarily, and an aesthetic color to show the range values. Using this graph empowers the detection of patterns across the countries and, also, makes it easy to see the time evolution. That way, an effective vision is created by having the countries on the $x$ axis and the index year on the $y$ axis. An example of this graph is shown in Figure 4c, where the countries are ordered alphabetically in the same line and with the same size.

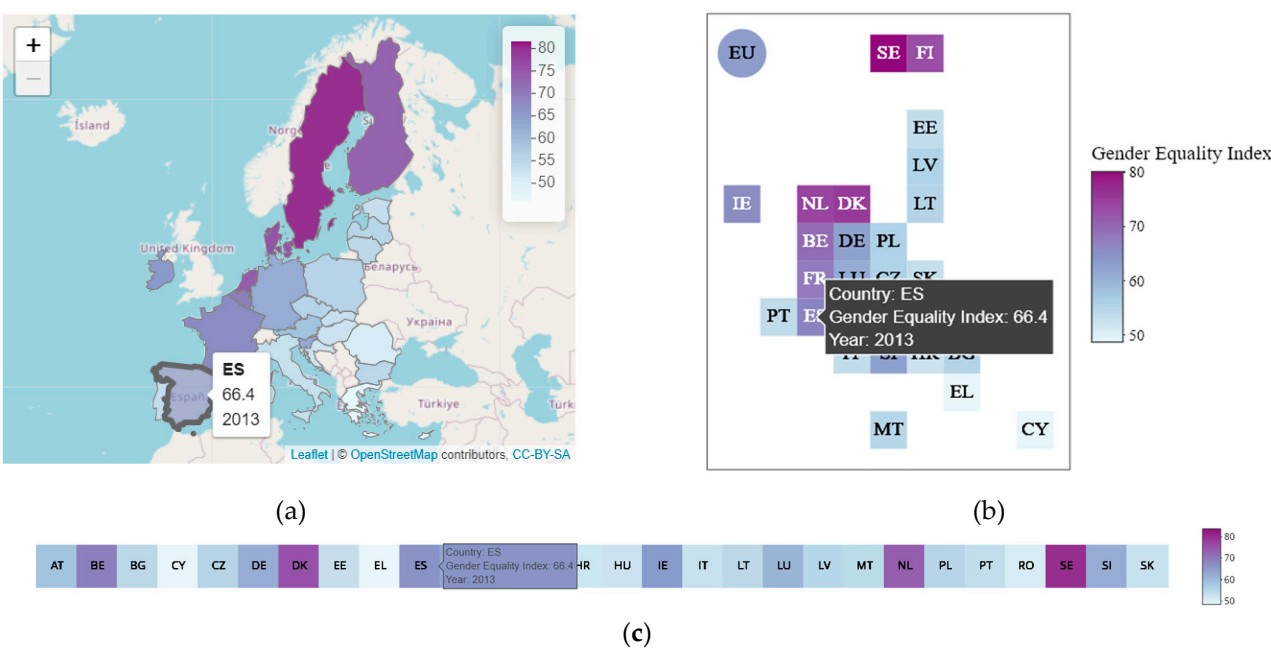

**Figure 4.** Visualizations to display the Gender Equality Index in 2013: (**a**) choropleth map; (**b**) cartogram–heatmap; (**c**) heatmap. Note that a pop-up appears in the three graphs on ES as the result of an action by the user (click on the graphical shape representing Spain).

### 3.3.2. Strategies

We have considered the following strategies to allow users a better experience when comparing different countries:

- Grids to compare all the years simultaneously or all the domains simultaneously: One of the essential problems of the EIGE visualizations is the lack of the possibility to compare the countries' changes over time side by side. By creating a 3 × 3 grid where each cell represents a year, it can be seen how the countries' indices have changed over time. As an example, Figure 5a shows a snapshot with a 3 × 2 grid, composed of the cartograms depicting the GEI in different years, from 2013 to 2021.

- Selection of two specific years, on dropdowns: Another potential way to compare different time periods is selecting a baseline year and another one to compare how much the countries have improved or become worse. This allows the user to determine which years to compare (consecutive years, first and last year, etc.) and identify patterns in the data. Figure 5b,c show the dropdowns to select the years (2013–2022) to calculate the differences, and the results are displayed in the form of a map.

- Relative vs absolute values for the color scale: To improve even further the last strategy, the scale is a key item. Since each domain has its own minimum and maximum values, different scales can be applied to the color ramp (min, max values) to better see the changes. We call this option the "relative" scale. On the other hand, if the purpose is comparing the changes between different domains (i.e., which domain has changed the most or the least, for two selected years), the same color ramp needs to be applied for all domains. We call this option the "absolute" scale. As an example, Figure 5b,c show the changes between 2013 and 2022 for the domain Time, depicted in form of a map. It can be seen that Figure 5b shows more variation in color; this is due to the fact that the Time domain between the selected years is the one with the least variations, which is evidenced when depicted as an absolute scale.

- Animated graphs: To understand better how the countries' GEI or domains vary over time, we think it is interesting to offer the users the ability to see how the values change in a sequential way by adding some animations. Thus, if the value changes, the color will change too, and users can identify them quickly. An example is displayed in

Figure 5d, where the values for four years are shown on a cartogram, with a slider to see the evolution.

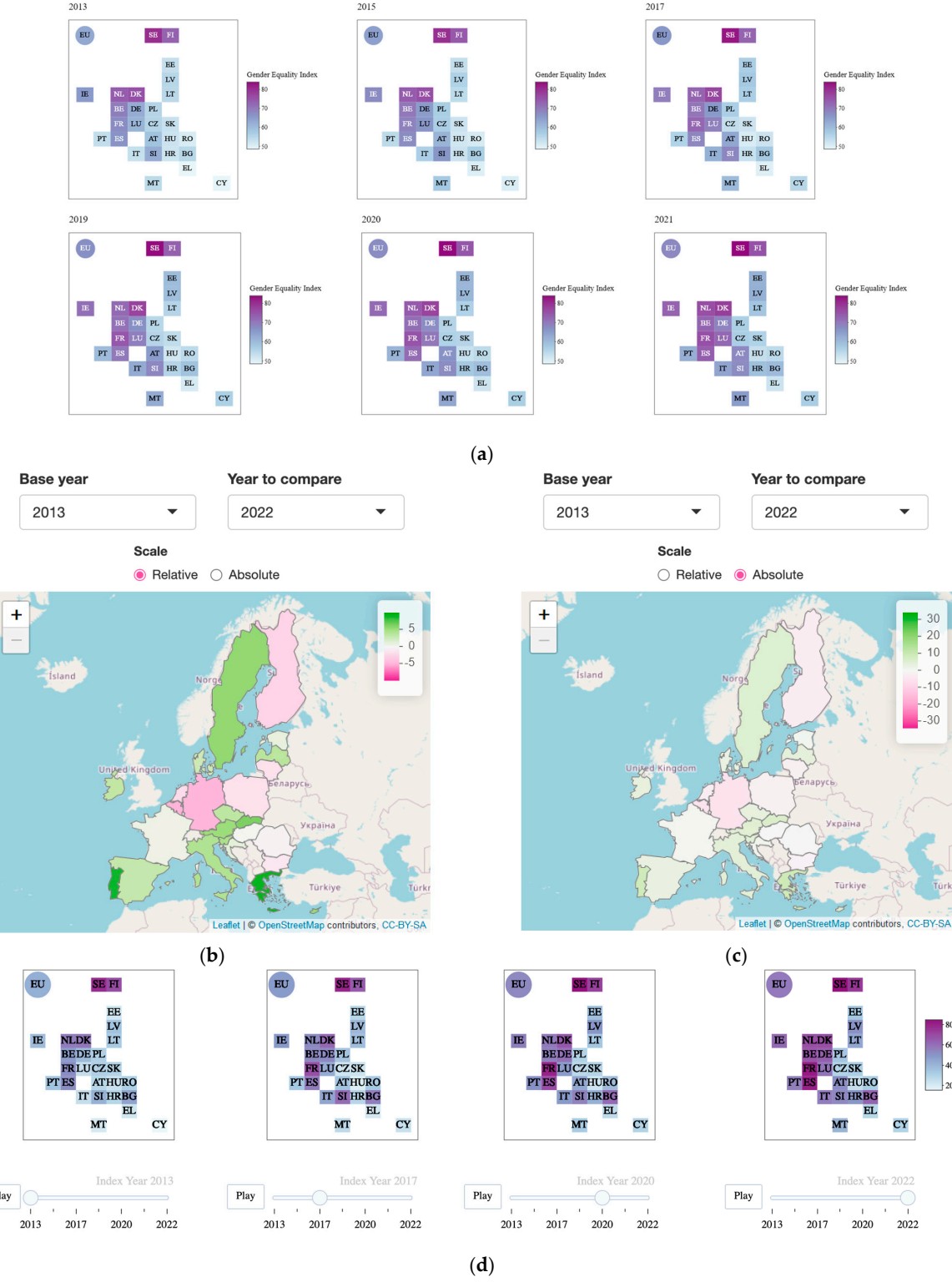

**Figure 5.** Strategies applied to different domains: (**a**) a 3 × 2 grid showing cartograms with the changes in the GEI from the year 2013 to the year 2021; (**b**) a map showing the changes from 2013 to 2022 for the domain Time, with a relative scale; (**c**) the same map as in "(**b**)", but in an absolute scale; (**d**) animated cartogram for the Power domain, showing the sequence of the years 2013, 2017, 2020 and 2022.

## 4. Design and Implementation

In this section, we will explain the design and implementation of an application that embeds a combination of the graphs and strategies mentioned above to help enhance the experience of users analyzing and comparing the GEI data of different countries and time periods. The application can be accessed from [56], and the code is openly available at a GitHub repository [57].

### 4.1. Design of the Graphical User Interface

Once you open the application, you can see the information tab where the purpose of the project is explained, as well as some useful links, such as the original data, the EIGE Gender Equality Index webpage or the DINA webpage.

There are also three tabs that allow the user to navigate through the different sections and visualize the data with a variety of visualizations. Each tab has the same options to choose the visualization: map, cartogram and heatmap.

First, there is the comparative tab (Figure 6). This tab allows the user to visualize the information corresponding to the selected category over time.

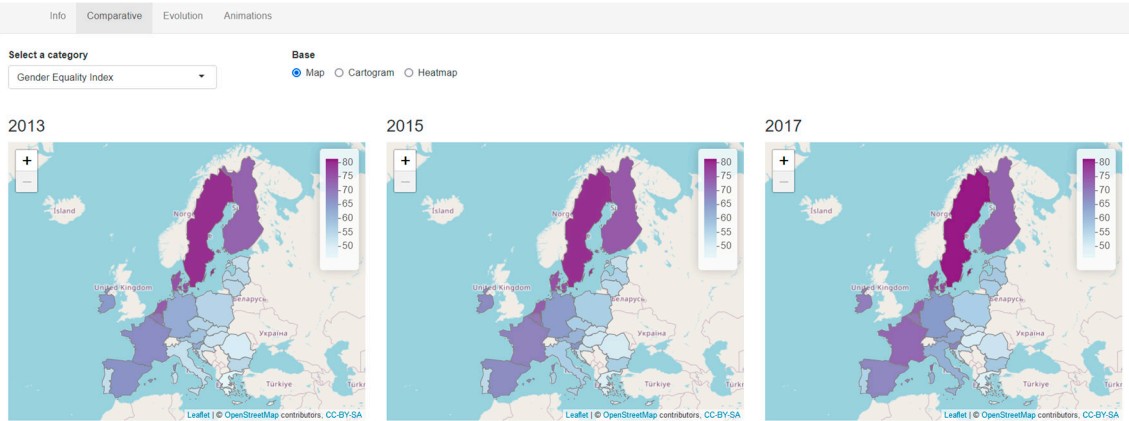

**Figure 6.** A snapshot of the Comparative tab that displays the corresponding category thought the years using maps.

On the second tab (Figure 7), the user can choose two years to compare the evolution of each domain. There is also a radio button that determines the scale of the visualizations: relative to each domain or absolute to see the general growth or decline.

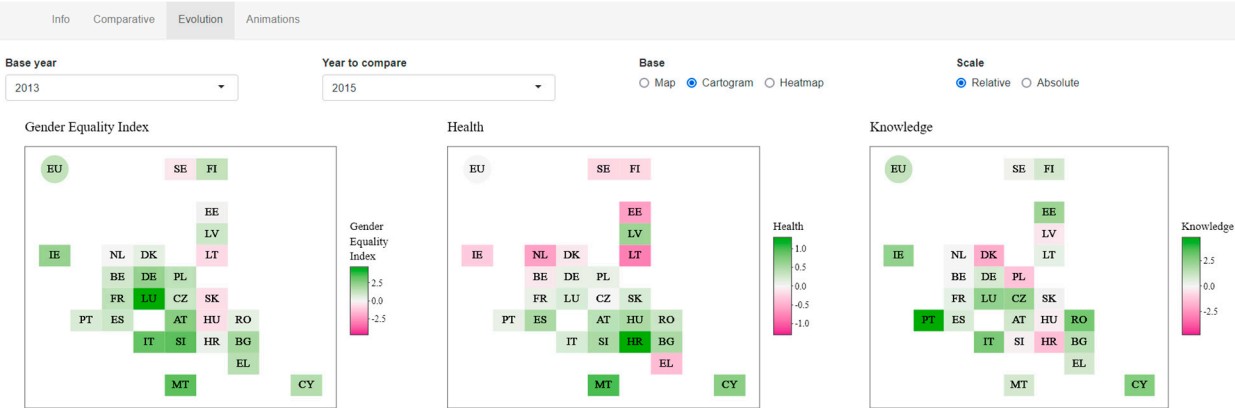

**Figure 7.** A snapshot of the Evolution tab that shows how the countries had evolved in a specific period (2013 and 2015) using cartograms and a relative scale.

Finally, the "Animation" tab (Figure 8) has a timeline of each domain, so the user can see the changes and identify quickly patterns in each country. This section is helpful

because, as the years passes, if the value changes significantly, there would be a change in the color.

**Figure 8.** A snapshot of the Animation tab where the user can see the changes of each category through animations using maps.

When the user is visualizing the map, there is a slider that can be activated to see the evolution of all the domains. In the case of the cartogram and heatmap, each domain is independent of the others, and there is a play button for each one.

### 4.2. Implementation of the Visualizations

The interface and the graphs have been developed making use of the R programming language and libraries, such as *shiny*, *leaflet* and *plotly*. To achieve these visualizations, the R package *ggplot2* has been used along with *dplyr* to achieve the filtered data, depending on the index year, the domain or the countries. This last case is referred to the decision of showing the UE value or not. The filters menus were made with dropdowns (domain, base year and year to compare) and radio buttons (base visualization and scale).

Moreover, since the application is accessible as open access, aimed to reach wide audiences, the color palettes selected are colorblind safe. That decision was made after evaluating those other gradient colors, such as red to green (including yellow as the center color) could cause problems for some people to discern between colors. Figure 9 shows the different sequential (Figure 9a) and diverging (Figure 9b) palettes that provide *ColorBrewer*, which are colorblind safe [58]. The gradients that we have used are (1) "BuPu" for the values of the index or domains, which is a sequential scale that goes from light blue to dark purple, and (2) "PiYG" for the evolution tab, which is a diverging scale that has on the extremes the pink and the green colors, and white for the midpoint color. A diverging scale has been chosen to represent decreasing and growing values, as this is an effective way to visualize the deviation of data values in one of two directions relative to a neutral midpoint [43].

As it was shown in Figure 4, most of the visualizations are interactive and display information, such as the country code, the domains and its value, once the user hovers over the different countries. This implementation was made thanks to libraries like *plotly*, which allows you to add interactivity to the existing graphs. Each graph has also some options: downloading the current graph, zooming in or out, selecting a certain part of the graph, etc.

#### 4.2.1. Maps

Before plotting any data on the map, a geographical shape of the countries is needed because the dataset from EIGE does not have this information. The shapefile has been downloaded from Eurostat [59] and edited using spatial analysis in QGIS to obtain only the countries of the EU27.

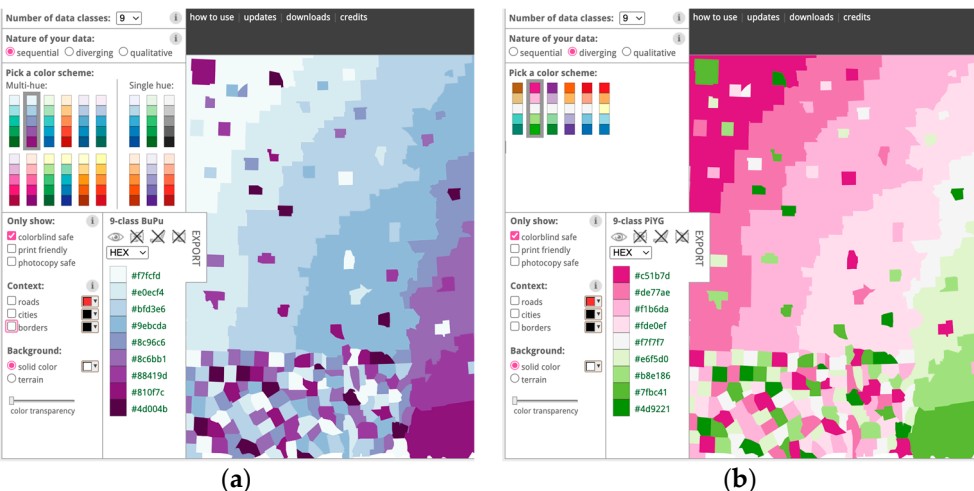

**Figure 9.** Colorblind-safe palettes from *RColorBrewer* [58]: (**a**) sequential scales, focusing on BuPu; (**b**) diverging scales, focusing on PiYG.

The base map is composed of tiles from OpenStreetMap, with the minimum zoom set to 3 to view all the countries. Then, the polygons are added using the value over a palette from the package *RColorBrewer* with highlight options, so the users can distinguish the borders of the countries when hovering over them. Finally, a pop-up appears when the mouse is in a country, and the country's code, the value of the index or domain and the year are shown.

### 4.2.2. Cartograms

To build cartograms, or more precisely, cartogram heatmaps, we explored different R packages. We found a R package called *statebins* that creates this type of cartogram, but it is specific to the United States. On the other hand, the *geofacet* package allows using some available grids or creating custom ones. With this package, we obtained different grid representations, which are shown in Figure 10. However, such representations were not entirely satisfactory, as they did not much maintain the topological relationships between countries. For instance, the grid in Figure 10a does not represent the real borders well: Malta (MT) does not have land borders with Italy (IT) and neither does Greece (EL) with Cyprus (CY) nor Italy (IT) with Luxemburg (LX). Meanwhile, Figure 10b solved some errors, but introduced new ones: Romania (RO) does not have a border with Lithuania (LT).

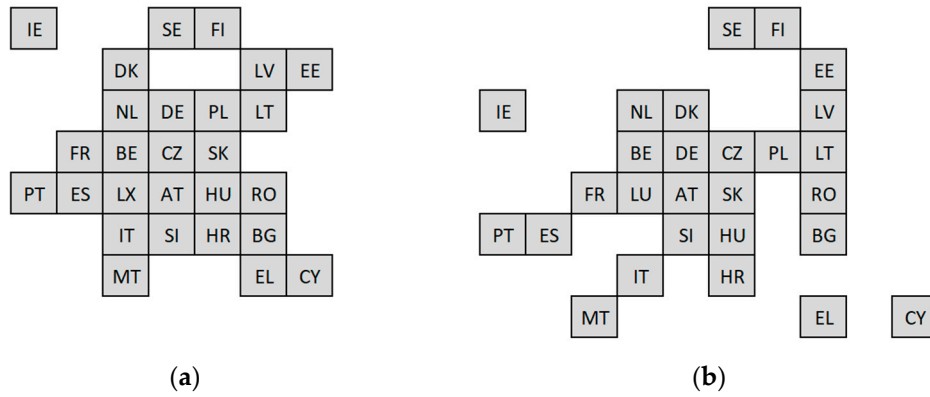

**Figure 10.** Available grids in geofacet R package displaying the countries of the European Union: (**a**) "eu_grid1"; (**b**) "europe_countries_grid1".

Hence, we decided to implement our own representation of a grid for the European Union (Figure 11), which solves those errors and maintains the borders in the best possible

way—though not a perfect solution exists with this basic representation—so the topological relationships are better preserved.

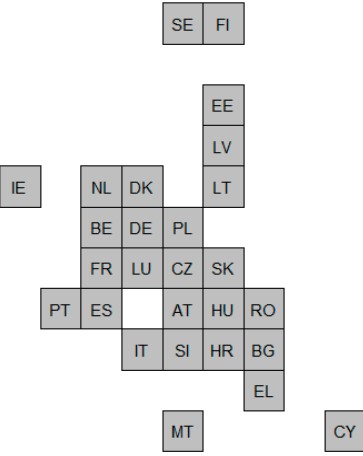

**Figure 11.** The proposed grid to represent the EU countries, based on [13].

To create this grid, each country was given a latitude and longitude number according to its location. When assigning the latitude ($y$ axis), countries to the south, like Malta or Cyprus, had smaller numbers, playing the same role as parallels. And the same thing was applied to the longitude with the meridians. The result is a $9 \times 11$ grid, where all the countries respect their borders and neighbors' countries.

The elements applied to this visualization are polygons to display the countries and points to display the score of the EU. Moreover, the code that identifies the country with two letters has been added to both elements and some scales: fill to display the value of the index or domains and color for the code of the countries. Finally, on each country, a pop-up with the same information as in the maps is included.

It Is worth mentioning that hexagons shapes are sometimes used in cartogram heatmaps instead of rectangles; to the best of our knowledge, clear advantages of using one or the other shape have not been formulated so far [55]. Instead, the best solution depends on the specific geography visualized and the task to be performed [54].

### 4.2.3. Heatmaps

We have considered two implementations for the heatmaps. The first one was created using the function *geom_bin2d*, resulting in a $7 \times 28$ matrix for each domain, where each tile represents a country in a specific year. The second one, a $1 \times 28$ matrix by each domain, was created with points elements transformed into squares for assigning the corresponding shape number. Both use gradient color ramps from *RColorBrewer*, with the legend on the right displaying the range and the values.

The first implementation allows users to compare the value of each domain for each country and time. Therefore, the countries codes are displayed on the $x$ axis, while the $y$ axis contain the years, removing the titles from the axis to show a cleaner graph.

The second one has the countries codes align on the $x$ axis, either using the axis text or annotated text over the rectangles. Since the size of the matrix is longwise, the heatmap has been created using a horizontal style rather than vertical to save space.

## 5. Results

In this section, we provide two examples to demonstrate how our proposed visualizations can complement the ones provided by EIGE. To that end, we provide some hypothetical questions, and then explore how they can be responded to, if possible, first taking into account EIGE's graphs, and then addressing the benefits of our visualizations.

### 5.1. Spain Evolution

As a first case study, we will try to respond to these questions: (1) How is the GEI for Spain evolving on the different domains? (2) Is this evolution better or worse than in other countries? (3) Does this evolution follow a similar pattern as that in the countries that are neighbors to Spain, i.e., Portugal and France?

In the "compare countries" tab, the EIGE allows visualizing the data within a choropleth map to compare the EU countries, as depicted in Figure 12. This figure shows the interface and the filters menu to choose a domain or subdomain and select the countries to compare.

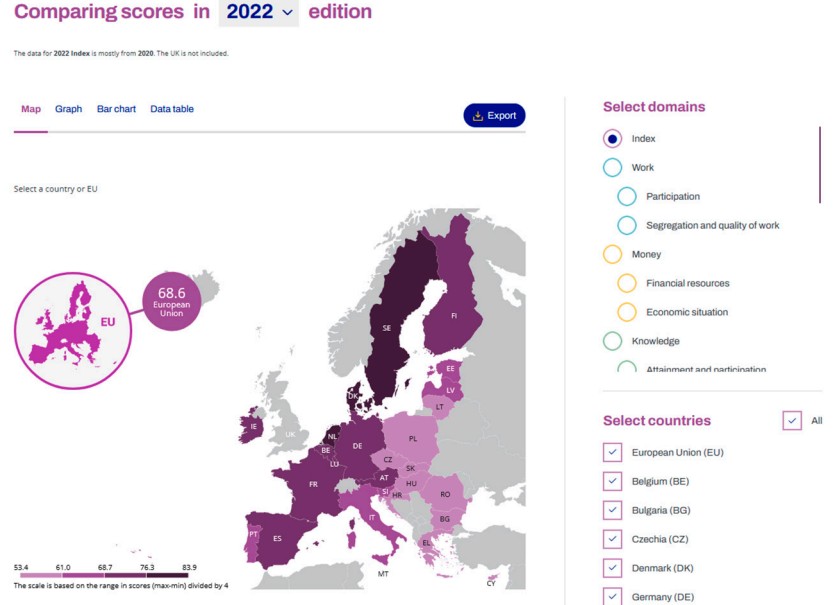

**Figure 12.** A snapshot taken from the EIGE's webpage, where the GEI for the year 2022 is shown, using a map to display the values of each country.

In this visualization, it is easy to compare the GEI value of Spain with the rest of the countries for the given year, but it is not possible to directly see their evolution over time. Although there is a dropdown to select the time periods, they are not comparable at a time. Thus, to try answering the given questions, one should explore the different years, one by one. Maybe the first question could be easily responded to, as the attention needs to be put only on one of the countries, i.e., one could see that the color of Spain becomes darker over the time, concluding that the evolution is positive. However, responding to the second of the questions means paying attention to other countries and their evolution, while selecting one year after another one, which makes it not possible to directly give an answer. The alternative selections of the radar chart, barplot (Figure 2) or data table (as provided in the "compare countries" tab) do not solve the problem, as they have the same restriction that only one year is seen at a time.

Now, let us revise the same data in the application that we have created, for instance, in the form of choropleth maps arranged in a grid, as depicted in Figure 13, where each map shows the GEI values for a year. Focusing attention on Spain, it can be easily seen that the value of the GEI has increased over time, as it turns darker. This would respond to the first question. To respond to the second question, Spain has been increasing the GEI, unlike the Eastern countries that have a similar value over the years. Finally, to respond to the third question, France and Portugal seem to have a similar evolution as Spain since their value also increases over the years. However, Portugal has lower GEI values than France.

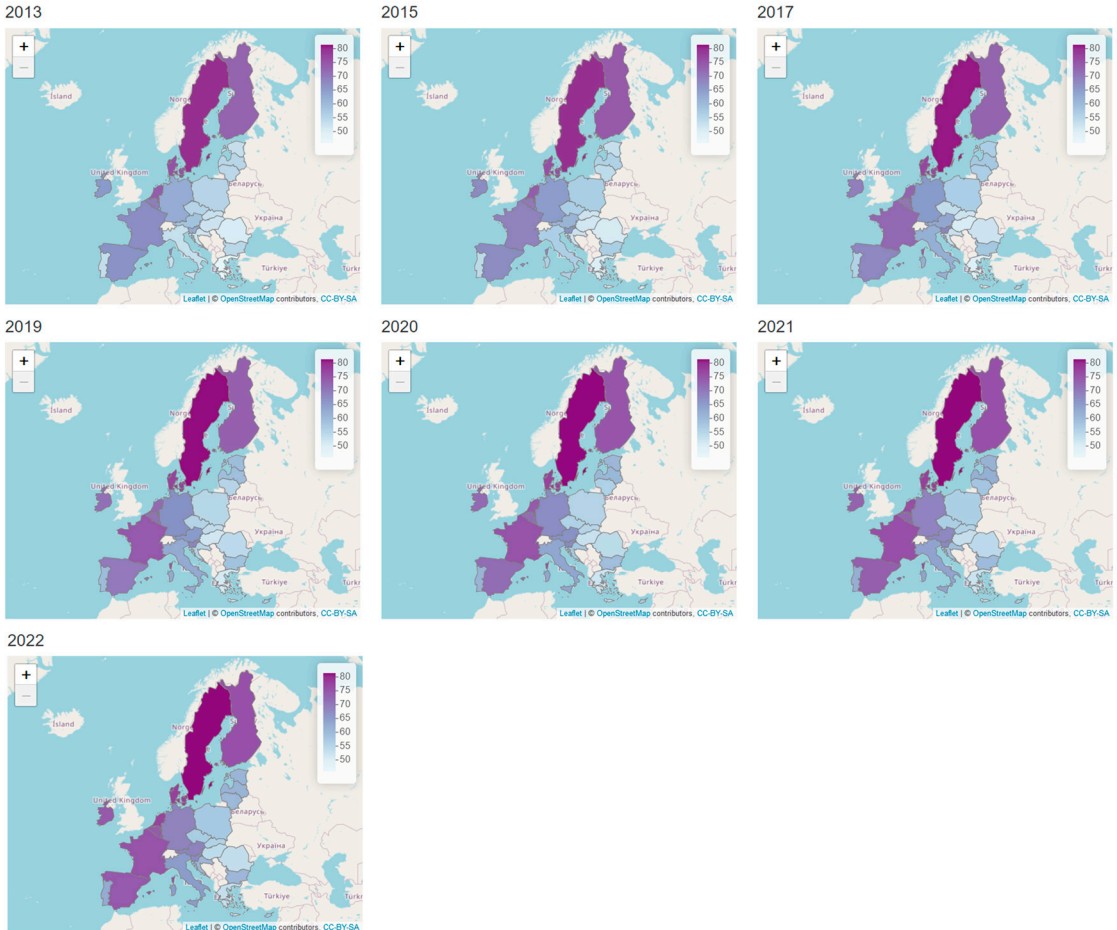

**Figure 13.** GEI over the years. Graph obtained from the Comparative tab.

On the other hand, the same questions can be explored for the different domains, using the same choropleth maps, but selecting a specific domain from the dropdown menu instead of the GEI.

Additionally, we can also try to respond to these questions, inspecting the evolution of two certain years by accessing the "Evolution" tab. With this option, we cannot directly see the behavior over time, but we can compare the starting and end period of times (2013 and 2022) to see if the indices are better or worse. Such an example is depicted in Figure 14. As it can be seen, Spain has increased in all domains, with the biggest growth in Power (28 points). In that domain, the evolution has been also positive for the rest of the countries, reaching the highest values for Italy (31.7 points). This process is not possible at the EIGE website, since you need to see and write down the values of the different years and then see the differences. However, it is straightforward in our application.

### 5.2. Smallest Countries in Europe

For this case study we will center our attention on the three countries of the EU with the smallest extension, which are Luxemburg (2586 km$^2$), Cyprus (9251 km$^2$) and Malta (316 km$^2$). We will try to respond to this question: Is the evolution of the GEI and domain indices similar for the three smallest countries in the EU?

Going back to the map displayed on the EIGE website (Figure 12), European countries with small surfaces are difficult to identify. It is truly remarkable the case of Malta, because on the map, one can only see the code of the country (MT), but not the color representing the value of GEI, even though when hovering over the country, a pop-up appears that shows the value. In those cases, it is necessary the use of zoom options to correctly visualize each country and see the area colored by the value.

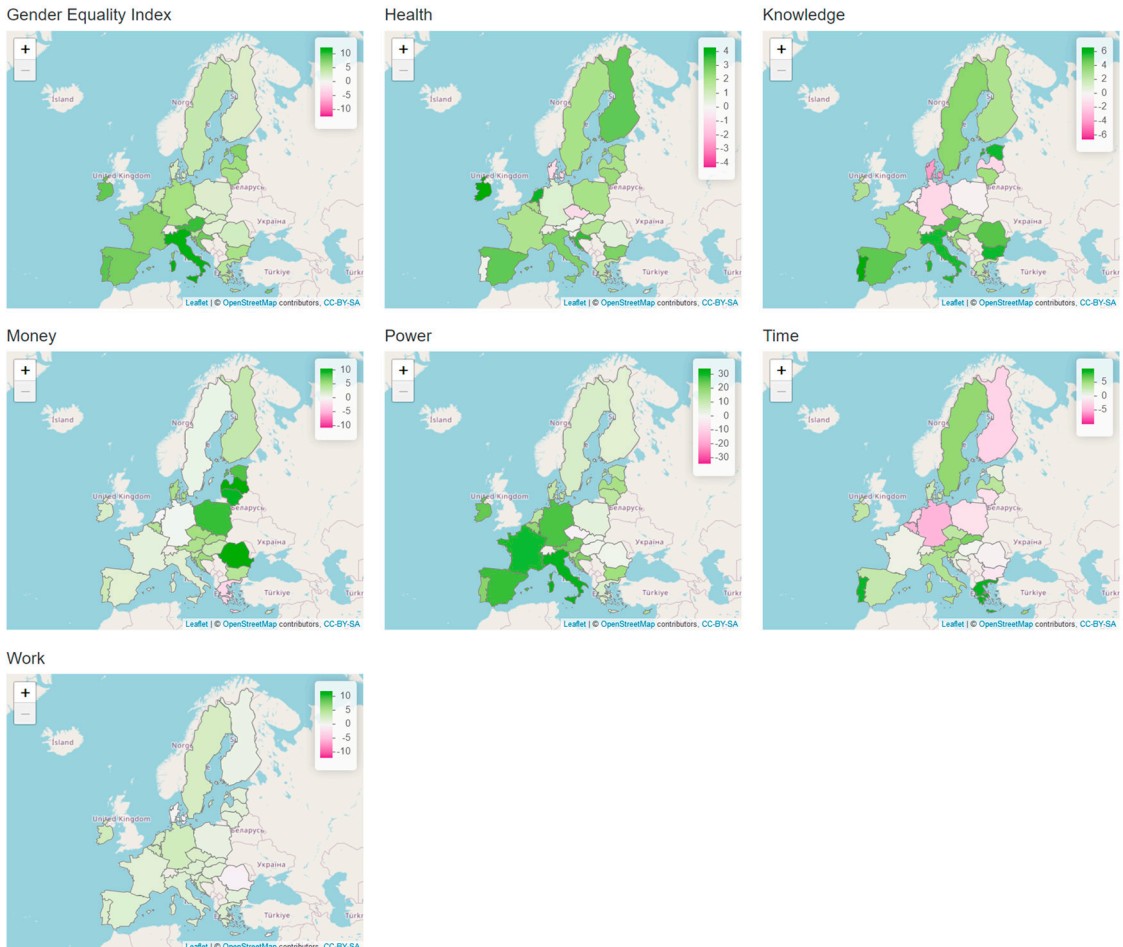

**Figure 14.** Changes of all the domains between 2013 (baseline year) and 2022, shown with a relative scale. Graph obtained from the Evolution tab.

Similarly, when using our proposed visualizations with the choropleth map, one should use the zoom to better see Luxemburg and Cyprus, or even to manage visualizing Malta. As an example, in Figure 15, the GEI map has been zoomed in (with the same level of zoom for the three images) to see the values and the corresponding colors of the three countries. This action takes a little time and, additionally, by doing that, the geographical reference to the rest of countries of the EU is lost.

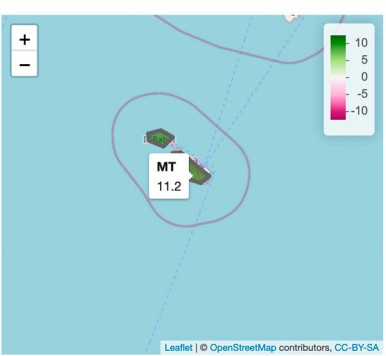 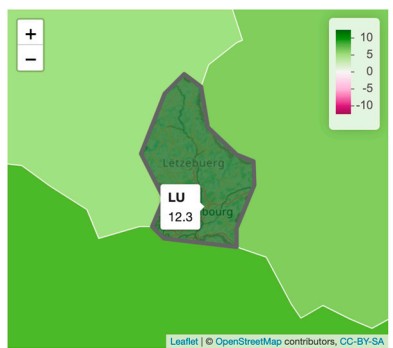 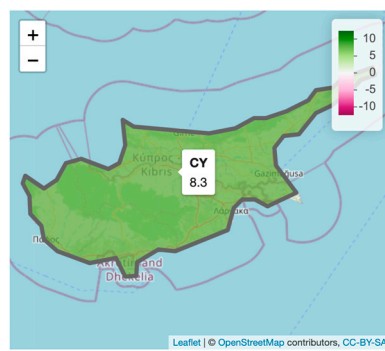

**Figure 15.** An example of the GEI changes (between the years 2013 and 2022) when zooming in on Malta (MT), Luxemburg (LU) and Cyprus (CY), by applying the same level of zoom. Graph obtained from the Evolution tab.

To solve those problems and be able to perfectly see each country and its value, we provide two options, to change the visualization to (1) a cartogram that shows the borders between countries or (2) a heatmap that shows the countries in a row in alphabetical order. And additionally, these graphs can be accessed from the different tabs, i.e., the Comparative, Evolution and Animation, each providing different strategies to visually inspect data. An example of the cartogram option accessed from the Evolution tab is seen in Figure 16, where we compare the changes from 2013 (baseline map) to 2022, with colors shown in a relative scale. In this case, the smallest countries are clearly depicted, while keeping the visualization of the rest of countries.

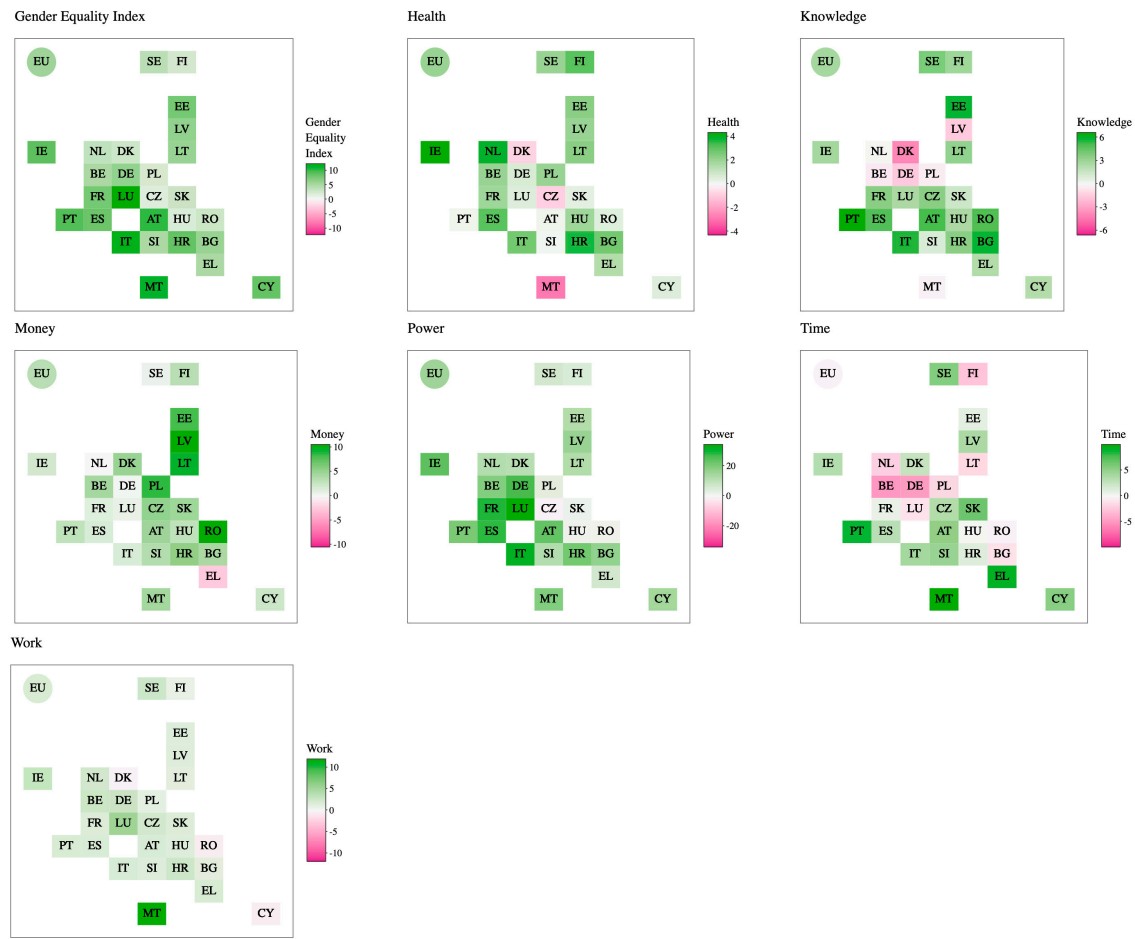

**Figure 16.** Changes in the indices of all the domains between 2013 (baseline year) and 2022, where each country is represented with the same size. Graph obtained from the Evolution tab.

With the cartogram, it is easier to answer to the question mentioned above. From 2013 to 2022, it can be seen that none of the three countries presents growth for all domains: Malta (MT) presents −2.8 points for Health and −0.2 in Knowledge, Cyprus (CY) −0.6 in Work and Luxemburg (LU) −1.1 in Time. For the rest of the domains (at each country, respectively), there is a growth in all the domains, and thus the GEI is positive for all.

But with the graphs of the Evolution tab, only the changes of two specific years can be directly compared. To compare the pattern of the evolution over the years, we can access the graphs of the Comparative tab to see the changes in a simultaneous way, or the Animation tab, to see the changes in a sequential way. As an example, in Figure 17, we show the heatmap in the Comparative tab for the domain Work, where a similar pattern of growth over the years can be seen for LU and MT as they increase progressively, though for MT, the changes are greater from one year to the next one; differently, CY presents a fluctuating behavior, with a shrink from 2013 to 2015, and a slightly lower value in 2022 than in 2013, as was also seen in Figure 16.

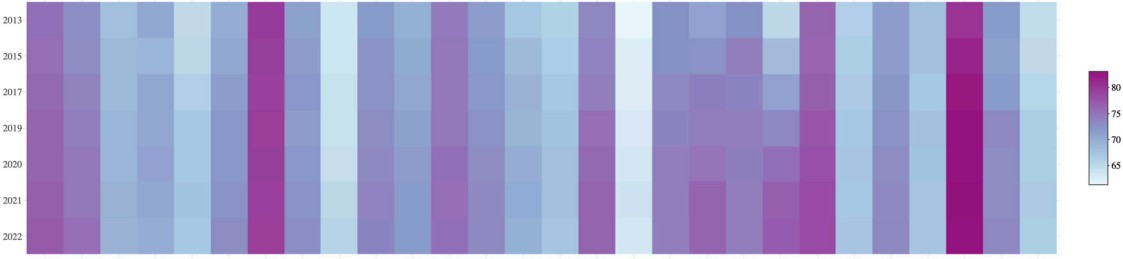

**Figure 17.** Changes in the indices of the Work domain over the years (from 2013 to 2022), seen as a heatmap. Graph obtained from the Comparative tab.

## 6. Discussion and Conclusions

Addressing gender inequalities in contemporary societies requires the availability of tools that can effectively visualize and make disparities visible. This article highlights the significance of employing data visualization techniques to inform the formulation of appropriate policies, thereby narrowing existing gender gaps and fostering a more egalitarian and just society.

This study demonstrates the potential of augmenting the analysis of data provided by the EIGE through innovative visualization methods. The presented work introduces a user-friendly, interactive tool that complements EIGE's existing data on the equality index, empowering the scientific community to conduct comprehensive studies and assess policy impacts over time and in specific contexts.

Despite the benefits of the proposed visualizations, a limitation of our tool is that not all the EIGE's available data are implemented in the visualizations, specifically, the original data have sub-domains (for each domain) that are not considered in the current graphical implementation. In this sense, it must be noted that our tool was conceived to complement EIGE's webpage, not to replace it. The sub-domain information is relevant when analyzing the raw data for each country, which can be consulted from the EIGE's webpage. However, an interesting further work for our tool would be to find visual strategies to allow integrating all the raw data provided by EIGE. In addition, as future work, we intend to integrate future index years as they are published by the EIGE, along with the possibility to predict the temporal evolution of future years, although this would require the consideration and study of sub-domains and the creation and study of temporal series with models, such as ARIMA (Auto Regressive Integrated Moving Average). Finally, we will schedule a comprehensive transnational usability test, in which we will not only try to measure user satisfaction, but also the degree of inclusiveness perceived by different groups of users with different social and cultural perspectives.

The obtained results indicate the feasibility of answering critical questions about temporary advancements in gender equality, which were previously unattainable with the available information from EIGE. The findings demonstrate the efficacy of the developed interactive tool in facilitating a deeper understanding of gender equality trends. By utilizing innovative data visualization, it becomes possible to gain valuable insights into the effectiveness of gender-related policies over time and within specific contexts. Consequently, this empowers policymakers and researchers with enhanced data to make well-informed decisions and design policies that foster gender equality. Further developments in this direction can lead to even more effective policymaking and sustained advancements in gender equality.

It should be noted that the result of this project is open access, accessible from [56].

**Author Contributions:** Conceptualization, Cristina Portalés; methodology, Cristina Portalés and Sergio Casas; software, Laya Targa and Cristina Portalés; validation, Laya Targa, Cristina Portalés, Sergio Casas, Jose Vicente Riera and Silvia Rueda; formal analysis, Laya Targa, Cristina Portalés, Sergio Casas, Jose Vicente Riera and Silvia Rueda; resources, Cristina Portalés; data curation, Laya Targa, Jose Vicente Riera and Cristina Portalés; writing—original draft preparation, Laya Targa, Silvia Rueda and Cristina Portalés; writing—review and editing, Laya Targa, Silvia Rueda, Sergio Casas, Jose Vicente Riera and Cristina Portalés; visualization, Laya Targa and Cristina Portalés; supervision, Cristina Portalés, Sergio Casas and Silvia Rueda; project administration, Cristina Portalés; funding acquisition, Cristina Portalés. All authors have read and agreed to the published version of the manuscript.

**Funding:** Cristina Portalés and L.T. are supported by the Spanish government postdoctoral grant Ramón y Cajal under grant No. RYC2018-025009-I.

**Data Availability Statement:** Publicly available datasets were analyzed in this study. This data can be found here: https://eige.europa.eu/modules/custom/eige_gei/app/content/downloads/gender-equality-index-2013-2015-2017-2019-2020-2021-2022.xlsx (accessed on 20 July 2023).

**Conflicts of Interest:** The authors declare no conflict of interest.

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
