# Peer review of "Enhancing the Understanding of the EU Gender Equality Index through Spatiotemporal Visualizations"

_ijgi, doi:10.3390/ijgi12100421_

Round 1

Reviewer 1 Report

- the reader don't get the information of how the research question was stablished. How did the authors identified the problem? Was only from your own experience using EIGE index?

- How have you assessed the use experience of people that works with the index?

- the solutions found were tested with users?

we have no information on all these questions.

It should be reviewed. In some points it is difficult to understand. (example lines 103-115)

Problems with some graphs (example 3.3.1 first point the reference to Fig 4 (a).

Reviewer 2 Report

The theme addressed in the text, is very relevant to the current context. The objectives of the text are clear. The methodological design is clearly described and substantiated and the results are clearly presented and discussed. However, it is considered that the theoretical foundation of the explored theme as well as the training methodology used is little developed and explored. Furthermore, the authors should describe in detail the different stages of the methodology used.

The proposal suggested by the authors of the text may allow research centered on gender issues in a systematic and comparable way, as it suggests a common unified framework.

It should be noted that the authors, when carefully selecting colors used in the different forms of visualization, sought not to promote discrimination themselves (eg people with color blindness).

Reviewer 3 Report

This work highlights the importance of using appropriate tools for data visualization, focusing on the realm of gender equality/inequality and the gender gap in European Union countries. Three types of graphs (maps, cartograms, and heat maps) are proposed, complementing the visualizations provided on the EIGE website, with an emphasis on spatial and temporal variables.

The authors are appreciated for their clear exposition. They have presented their work in a very clear and concise document, providing well-justified arguments for developing the aforementioned three types of graphs.

Regarding the text review, I have some observations which I outline below:

- A possible inconsistency has been detected in line 60. While 8 different domains are mentioned, only 6 are cited (indeed, throughout the document and on the EIGE website, 6 are mentioned).

- In the last paragraph of the first section (lines 116 to 122), the phrase "we provide" is used twice to introduce sections 3 and 4. The authors should replace one of them with a synonym.

- Furthermore, I believe that the third paragraph of the second section (lines 131 to 139) should be supported by a bibliographic reference.

- Figure 11 presents the authors' proposal for representing all EU countries with the same size. They use squares, but other options like hexagons used in similar graphs are not discussed. In fact, some authors argue that hexagons offer a representation closer to reality. The authors should provide more justification for their choice.

- The authors have meticulously presented the work they have carried out, indicating that it is available openly on the website https://server1.uv.es/GenderEquallyIndex/. However, they have not mentioned that the R code used is not available openly, at least on that website. They have also not commented on the delay in visualizing the graphs they have created and displayed on their website, a circumstance that is not observed with the graphs on the EIGE website.

In any case, I find the paper to be very interesting, and I believe it should be published, taking into account the aforementioned observations, in the ISPRS International Journal of Geo-Information.

Round 2

Reviewer 1 Report

I believe the paper has improved considerably and is now ready for publication.

Reviewer 3 Report

After reviewing the paper and noting that the authors have taken into account all the comments and suggestions proposed in the initial review, I believe that the work "Enhancing the Understanding of the EU Gender Equality Index Through Spatiotemporal Visualizations" should be published in the International Journal of Geo-Information.